# Cerebral blood flow and cerebrovascular reactivity are preserved in a mouse model of cerebral microvascular amyloidosis

Leon P Munting[1,2], Marc Derieppe[1,3], Ernst Suidgeest[1], Lydiane Hirschler[1], Matthias JP van Osch[1], Baudouin Denis de Senneville[4,5], Louise van der Weerd[1,2]*

[1]Department of Radiology, Leiden University Medical Center, Leiden, Netherlands; [2]Department of Human Genetics, Leiden University Medical Center, Leiden, Netherlands; [3]Princess Máxima Center for Pediatric Oncology, Utrecht, Netherlands; [4]Department of Radiotherapy, University Medical Center Utrecht, Utrecht, Netherlands; [5]Institut de Mathématiques de Bordeaux, Université Bordeaux/CNRS UMR 5251/INRIA, Bordeaux-Sud-Ouest, France

**Abstract** Impaired cerebrovascular function is an early biomarker for cerebral amyloid angiopathy (CAA), a neurovascular disease characterized by amyloid-β accumulation in the cerebral vasculature, leading to stroke and dementia. The transgenic Swedish Dutch Iowa (Tg-SwDI) mouse model develops cerebral microvascular amyloid-β deposits, but whether this leads to similar functional impairments is incompletely understood. We assessed cerebrovascular function longitudinally in Tg-SwDI mice with arterial spin labeling (ASL)-magnetic resonance imaging (MRI) and laser Doppler flowmetry (LDF) over the course of amyloid-β deposition. Unexpectedly, Tg-SwDI mice showed similar baseline perfusion and cerebrovascular reactivity estimates as age-matched wild-type control mice, irrespective of modality (ASL or LDF) or anesthesia (isoflurane or urethane and $\alpha$-chloralose). Hemodynamic changes were, however, observed as an effect of age and anesthesia. Our findings contradict earlier results obtained in the same model and question to what extent microvascular amyloidosis as seen in Tg-SwDI mice is representative of cerebrovascular dysfunction observed in CAA patients.

*For correspondence:
L.van_der_Weerd@lumc.nl

**Competing interests:** The authors declare that no competing interests exist.

## Introduction

Cerebral amyloid angiopathy (CAA) is a neurovascular disease characterized by accumulation of the amyloid-β peptide in the brain vasculature, which ultimately leads to stroke and cognitive decline (*Banerjee et al., 2017*). In both hereditary and sporadic variants of CAA, measurements in patients have shown that impairments in cerebrovascular function can be found early in the disease process (*Dumas et al., 2012*; *van Opstal et al., 2017*). These measurements were performed with blood oxygenation level dependent-functional magnetic resonance imaging (BOLD-fMRI) and a visual stimulation paradigm, allowing to characterize the cerebrovascular response to neuronal activation in the occipital cortex, where visual processing occurs. The occipital cortex is also where CAA burden is highest (*Yamada et al., 1987*), likely contributing to the sensitivity of the fMRI measurement.

Mouse models of cerebral amyloidosis are invaluable tools for testing safety and effectiveness of greatly needed novel therapies for amyloid-β-related diseases, including CAA. However, for the results to be translatable to the clinic, it is essential that the amyloidosis model shows similar structural and functional phenotypes as the patient. The transgenic Swedish Dutch Iowa (Tg-SwDI) mouse model is an amyloidosis model expressing low levels of the human amyloid-β precursor protein (APP) gene with three familial mutations, of which the Dutch and Iowa mutations are located in the amyloid-β coding region of APP (*Davis et al., 2004*). The neuronal expression of the mutated APP in

this model leads to early amyloid-β accumulation in the brain, starting around 6 months. Amyloid-β accumulates predominantly around capillaries in the Tg-SwDI model, which is similar to a subtype of CAA pathology observed in pathological examinations of patient tissue and sometimes referred to as capCAA (*Thal et al., 2002*). It is unknown to what extent amyloid-β accumulation around capillaries contributes to the observed impairments in cerebrovascular function in patients. Unlike patients, vascular pathology in the Tg-SwDI model is most severe in the thalamus (*Miao et al., 2005*). Previous studies have measured cortical vascular reactivity using laser Doppler flowmetry (LDF) after removal of the skull in Tg-SwDI and wild-type (WT) mice. They reported, similar to CAA patients, early impairments in cerebrovascular reactivity (CVR) in Tg-SwDI mice (*Chow et al., 2007*; *Park et al., 2014*). It is unknown, however, whether skull removal has affected the outcome, and whether the thalamus is more strongly affected, as the thalamus is not readily accessible with LDF.

Here, we therefore used the noninvasive arterial spin labeling (ASL)-magnetic resonance imaging (MRI) technique to study the cerebrovascular function in the Tg-SwDI brain. With ASL, arterial blood is magnetically labeled and used as endogenous tracer flowing into the tissue of interest, which is most often the brain. The distribution of the label over the different brain regions reflects local tissue perfusion and can be converted into absolute cerebral blood flow (CBF) values, expressed as mL/100 g/min. When combined with a hypercapnic challenge, both CBF and CVR can be determined for different brain regions. A known unfavorable characteristic of ASL is a possible underestimation of CBF in case of slow flow. In that case, a delayed arterial transit time (ATT) – the time that it takes for the label to travel from the labeling plane to the brain tissue – could be misinterpreted as decreased CBF. ATT evaluation is therefore valuable to prevent a potential underestimation of the ASL-based CBF estimates, as well as indicative in itself of vascular pathology (*Alsop et al., 2015*).

The noninvasive nature of ASL was fully exploited in our study with the use of a longitudinal study design in which CBF and CVR were repeatedly measured during increasing amyloid-β accumulation in the brain vasculature of Tg-SwDI mice. As it is conceivable that high microvascular amyloid burden could lead to delayed ATT in Tg-SwDI mice, an ATT measurement was added to the protocol by means of a modified ASL sequence optimized to capture the inflow of the tracer into the brain tissue (*Hirschler et al., 2018a*). To allow for repeated measurements, a minimally invasive isoflurane anesthesia protocol was used. Moreover, to be able to compare the results to literature, additional end-point measurements were performed under a terminal anesthesia protocol with urethane and α-chloralose (U and A). Furthermore, a subgroup of mice was used to directly compare ASL-MRI to LDF. The study design is summarized in *Figure 1*.

## Results

Hypercapnia consistently induced a CBF increase in both WT and Tg-SwDI mice under isoflurane anesthesia (*Figure 2* and *Figure 3*), which was mainly located in cortical regions (*Figure 2*). Surprisingly, the CBF maps and CBF time profiles acquired in the mid-brain of the Tg-SwDI mice resembled those of the WT mice at every time point. Hence no significant differences were observed in baseline CBF nor CVR between the two genotypes at any time point (*Figure 3b*; see *Figure 3—figure supplement 1* for individual animal trends). A significant effect of age on CBF was, however, observed in both WT and Tg-SwDI mice, $\chi^2(3)=13.00$, p=0.005 and $\chi^2(3)=8.49$, p=0.037, respectively. Post hoc analysis indicated that this was attributable to a decrease in baseline CBF between the ages of 3 and 6 months. Between these time points, median (iqr) CBF significantly decreased from 155 (143–159) to 121 (112–124) mL/100 g/min in WT mice, p=0.008. In Tg-SwDI mice, a similar trend was observed, from 147 (133–151) to 126 (103–135) mL/100 g/min, p=0.036, but this was not significant (cut-off p-value of 0.017 after Bonferroni correction). From 6 months of age, the baseline CBF remained stable inside each group. Age also had a significant effect on CVR in WT mice, $\chi^2(3)=8.33$, p=0.040. No differences were observed with post hoc analysis, however, besides a trend toward increased CVR between the ages of 3 and 6 months, from 13 (7–17) to 31 (23–37) %, p=0.038. Age had no significant effect on CVR in Tg-SwDI mice, $\chi^2(3)=6.94$, p=0.074. Additional analysis in cortical and thalamic areas did not reveal any difference between WT and Tg-SwDI mice either (*Figure 3— figure supplement 2*). Similarly, no differences were observed in brain volume, body weight, change in tc-pCO$_2$ upon the hypercapnia challenges, respiration rate, and inversion efficiency between WT and Tg-SwDI mice (*Figure 3—figure supplement 1*, *Figure 3—figure supplement 3* and *Figure 3—*

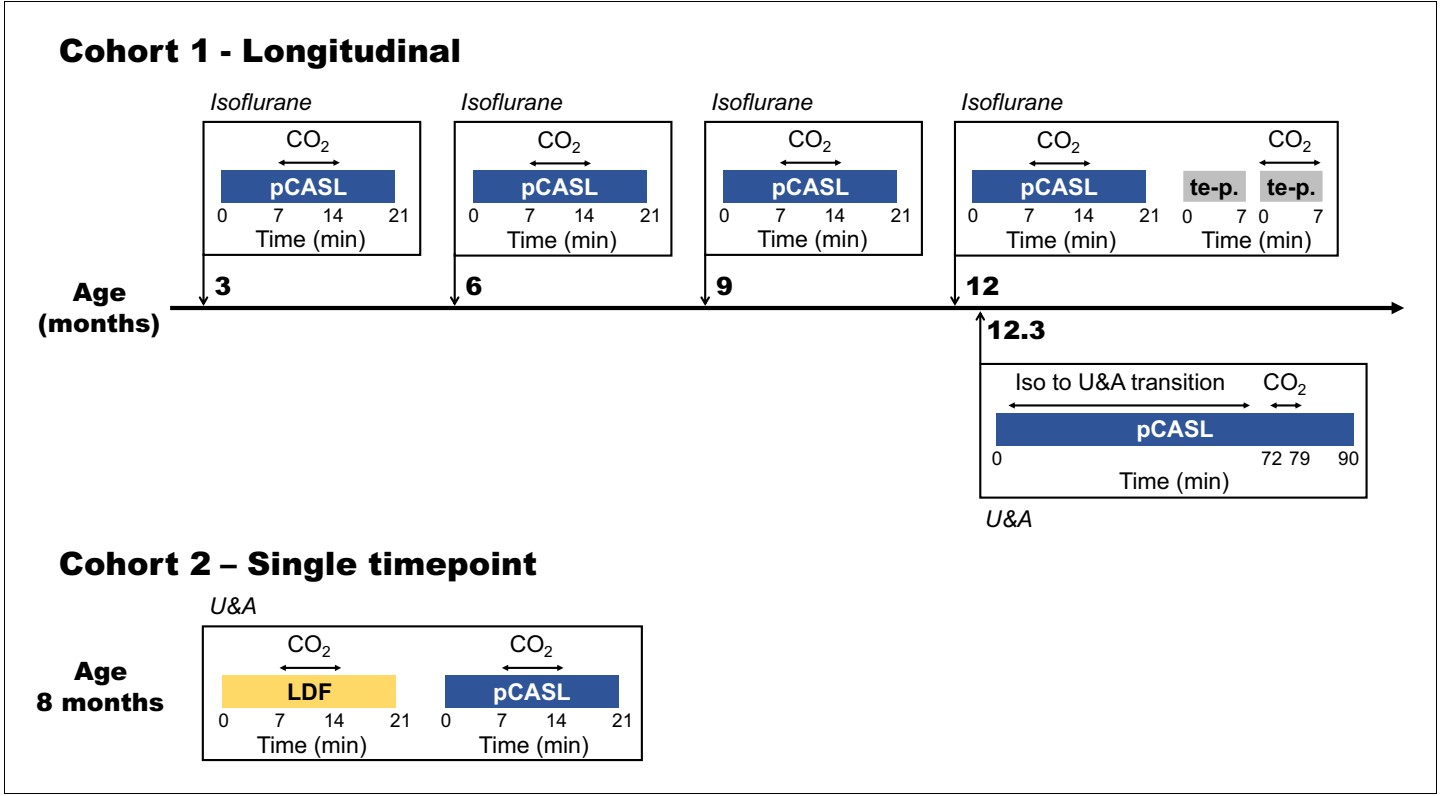

**Figure 1.** Study design. Two different cohorts were used in this study, of which the first was followed longitudinally. The timeline of the first cohort is illustrated in the upper part of the figure, with scan moments indicated with orthogonal arrows projected onto the time line. The most relevant scans performed at these moments are indicated within the boxes adjoined to the orthogonal arrows and the type of anesthesia used is indicated in italics on top of the boxes. The lower part of the figure illustrates the single time point measurements performed in cohort 2. pCASL = pseudo-continuous arterial spin labeling; te p. = time-encoded pseudo-continuous arterial spin labeling; U and A = urethane and α-chloralose; LDF = laser Doppler flowmetry.

*figure supplement 4*). Of note, the CBF at 12 months showed higher variability in the WT group, and the body weights showed higher variability in the Tg-SwDI group.

Median (iqr) baseline ATT values in the mid-brain were also similar for both WT and Tg-SwDI mice, i.e. 206 (186-240) milliseconds (ms) and 223 (202–246) ms respectively at 12 months of age (*Figure 4*). The hypercapnia challenge shortened the ATT to 192 (189–205) for WT and 197 (187–207) for Tg-SwDI mice, which was significant for Tg-SwDI mice (Z = −2.20 and p=0.028), but did not reach significance for WT mice (Z = −1.84 and p=0.066).

To evaluate if a functional deficit could have been masked by the vasodilatory effect of isoflurane during the MRI sessions, additional CBF and CVR measurements were performed in the same cohort of mice under urethane and alpha-chloralose (U and A) anesthesia. This was done at 12.3 months, 10 days after the last MRI measurement under isoflurane. The change of anesthesia protocol resulted in profound hemodynamic changes: baseline CBF was markedly reduced and the hypercapnic response was higher in amplitude, but also slower (*Figure 5a*), and more widespread in the brain tissue (*Figure 5b*). The CBF and CVR estimates were indeed significantly impacted by the change in anesthesia protocol (*Figure 5c*), with the median CBF (iqr) decreasing from 126 (84–141) to 28 (26–30) mL/100 g/min in WT mice, Z = −2.67 and p=0.008, and median (iqr) CVR increasing from 26 (6–47) to 233 (193–245) %, Z = −2.67 and p=0.008. These changes were again comparable to those in Tg-SwDI mice, with the median CBF (iqr) and CVR (iqr) respectively changing from 114 (103–130) to 25 (21–40) mL/100 g/min, Z = −2.37 and p=0.018, and from 30 (17–34) to 265 (178–312) %, Z = −2.37 and p=0.018. The CBF response at the induction phase of U and A anesthesia was also similar in Tg-SwDI and WT mice (*Figure 5—figure supplement 1*). The higher CVR during U and A was unlikely due to a higher $CO_2$ absorption, as the tc-p$CO_2$ responses to the hypercapnia challenges only

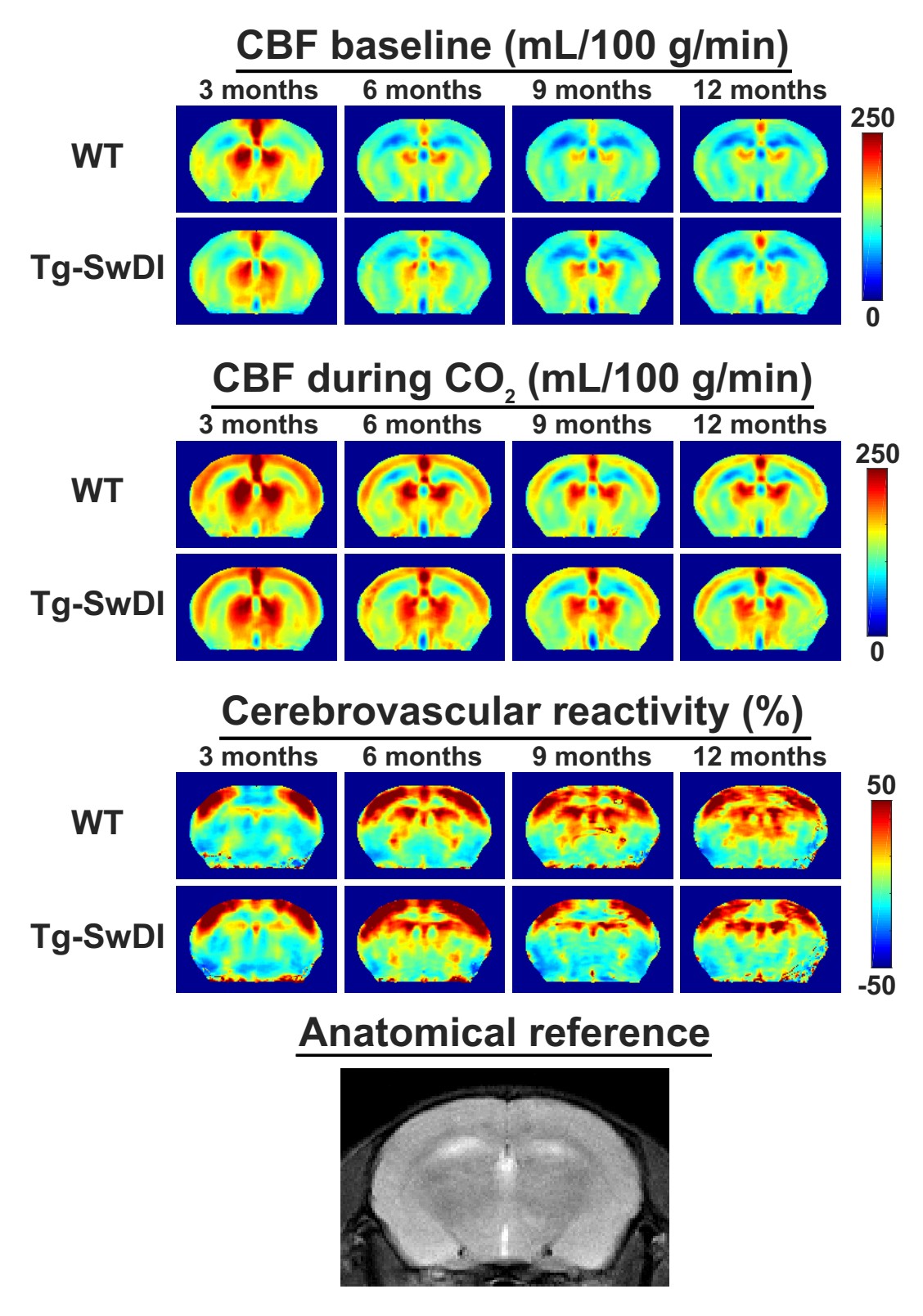

**Figure 2.** Average mid-brain cerebral blood flow (CBF) and cerebrovascular reactivity (CVR) maps for wild-type (WT) and transgenic Swedish Dutch Iowa (Tg-SwDI) mice in cohort 1. From left to right, the different ages are displayed. From top to bottom, respectively CBF maps at baseline, CBF maps during $CO_2$, and CVR maps are displayed, with WT and Tg-SwDI mice alternating per row. On the bottom row, an anatomical magnetic resonance

*Figure 2 continued on next page*

*Figure 2 continued*

imaging (MRI) scan of the same brain slice is shown. Note that the CBF increase during the $CO_2$ challenge is most profound in the cortex, and that the WT and Tg-SwDI mice show similar CBF and CVR maps.

increased from an average of 15 mmHG during isoflurane to an average of 19 mmHG during U and A (*Figure 5—figure supplement 2*).

An additional smaller second cohort of mice was used to cross-validate our MRI findings with the previously used LDF readout as imaging modality to assess cerebrovascular function in this mouse model (*Chow et al., 2007*; *Park et al., 2014*). After a unilateral craniotomy (right side), LDF measurements were performed with two probes at the same time: one through the skull in the left hemisphere and the other directly above the brain tissue in the right hemisphere. MRI measurements were performed directly after the LDF measurement in the same mice and the brain region analyzed with the MRI data was restricted to the somatosensory cortex, where the LDF measurements were also collected. No differences in CVR could be observed between WT and Tg-SwDI mice, neither with MRI nor with LDF (*Figure 6*). Of note, removal of the skull severely reduced the CVR for both imaging modalities (*Figure 6—figure supplement 1*).

Lastly, the brain tissue was stained for amyloid-β to assess the degree of pathological burden. All Tg-SwDI mice developed extensive amyloid-β plaque pathology by the end of the experiment, with mainly diffuse parenchymal plaques in the cortex and microvascular plaques in the hippocampus and thalamus, but none of the WT mice displayed any amyloid-β deposition (*Figure 7*).

## Discussion

This longitudinal study characterized cerebrovascular function in the Tg-SwDI mouse model of microvascular amyloidosis over the full course of amyloid-β pathogenesis. No significant impairment in cerebrovascular function could be found in the Tg-SwDI model, no matter the age or functional parameter explored. This contradicts previous findings, which showed early impairments in CVR in the cerebral cortex in Tg-SwDI mice using LDF (*Chow et al., 2007*; *Park et al., 2014*). In our study, cerebrovascular function was first assessed using ASL-MRI and a $CO_2$ challenge, allowing to measure absolute CBF as well as CVR in the entire brain. The ASL-MRI perfusion data were acquired longitudinally using a low-level isoflurane anesthesia protocol to capture the dynamics of decreasing cerebrovascular function over increasing microvascular amyloid-β loads. This study setup was substantially different from literature studies, where relative CBF measurements performed using LDF under a terminal anesthesia protocol with urethane and α-chloralose (U and A), after removal of the skull, showed impaired hemodynamics in the Tg-SwDI mouse model (*Chow et al., 2007*; *Park et al., 2014*). Therefore, additional experiments were performed to determine whether differences in experimental design could explain this discrepancy. A likely candidate was the difference in anesthesia protocol, as this has been shown to significantly influence CVR experiments (*Petrinovic et al., 2016*; *Munting et al., 2019*). Another likely candidate was the difference in imaging modality, as ASL-MRI and LDF are sensitive to different blood components, namely flux of blood plasma (ASL-MRI) or velocity of red blood cells (LDF). However, after differences in anesthesia protocol and imaging modality were accounted for, cerebrovascular function was in our hands still found to be preserved in the Tg-SwDI model. Practically, it is very difficult to replicate an experiment up to the smallest detail in a different laboratory, as small differences might remain. However, by showing similar findings for two modalities and two anesthesia protocols, including the ones used before, we think that our study convincingly shows that microvascular amyloidosis in the Tg-SwDI mouse model does not induce cerebrovascular dysfunction. Moreover, the robustness to sense local hemodynamic changes could be confirmed based on the observed reduced CBF and CVR as a function of age and anesthesia. The CBF time profiles of Tg-SwDI mice were in fact remarkably similar to their WT controls, even when CBF was monitored for up to 1.5 hr (*Figure 5—figure supplement 1*). Everything considered, our functional results indicate that the causal link between microvascular amyloidosis and cerebrovascular function, which was established in past studies in the Tg-SwDI model, is to be mitigated and remains to be fully uncovered.

Some remaining differences between our approach and those of others are, however, useful to mention and could possibly provide explanations for the different outcomes found between studies.

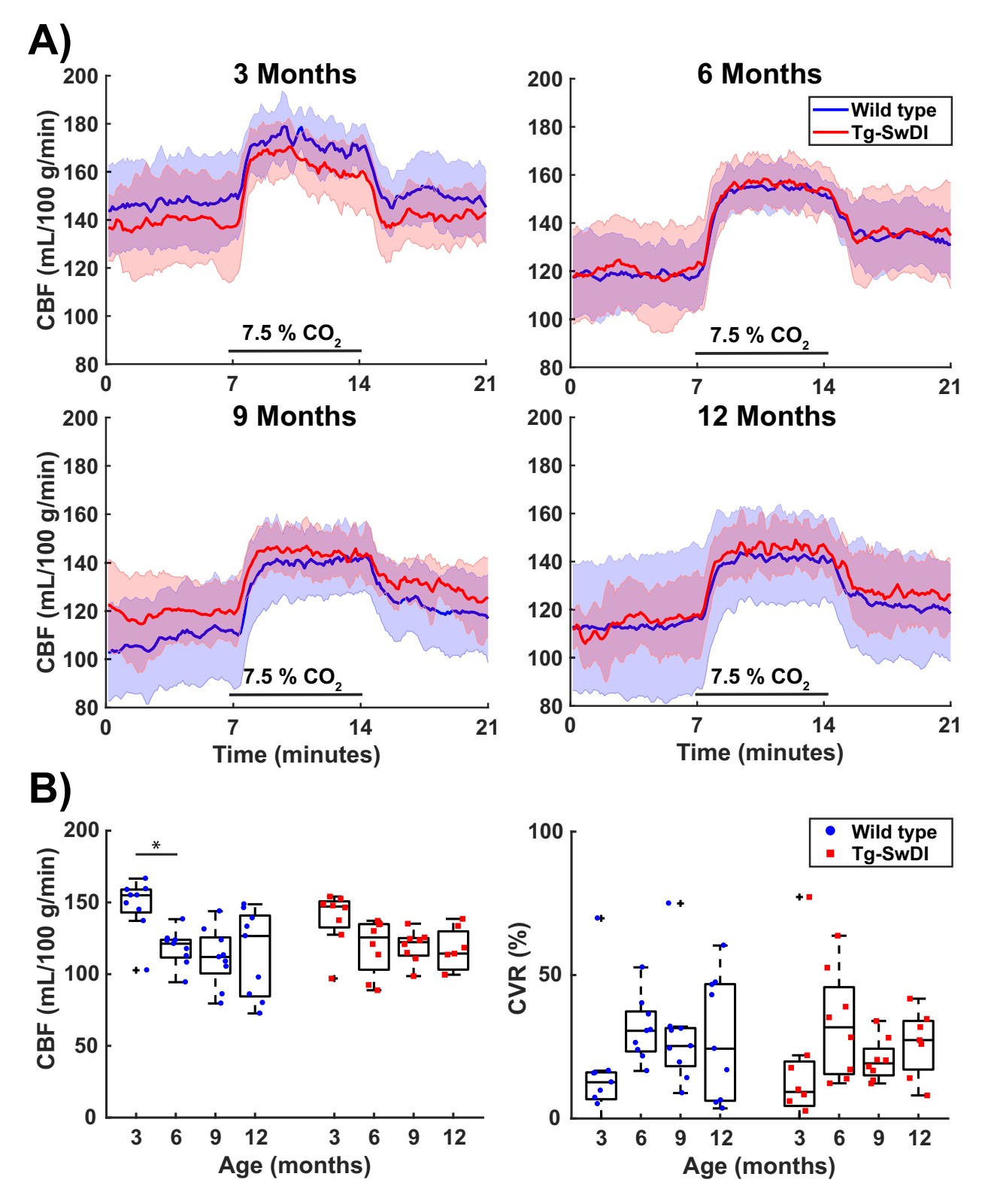

**Figure 3.** Cerebral blood flow (CBF) and cerebrovascular reactivity (CVR) values acquired in the mid-brain in cohort 1. (A) 21-min CBF time profiles (mean ± standard deviation) that were retrieved in a full mid-brain slice at the ages of 3, 6, 9, and 12 months old shows for wild-type (WT) and transgenic Swedish Dutch Iowa (Tg-SwDI) mice. $CO_2$ was administered between minutes 7 and 14. (B) Boxplot representations of baseline CBF (average of the last 2.3 min before the start of $CO_2$ administration) and CVR (ratio of average of the last 2.3 min during $CO_2$ to baseline CBF). Circles and

*Figure 3 continued*

squares represent individual mice. No significant differences were observed between the two genotypes, but there was a significant effect of age (Friedman test, p=0.005 for CBF in WT, p=0.037 for CBF in TG, p=0.040 for CVR in WT). From the post hoc analysis, only the drop in CBF in WT mice between 3 and 6 months old (p=0.008) reached the Bonferroni-corrected significance threshold (p=0.017).

The online version of this article includes the following figure supplement(s) for figure 3:

**Figure supplement 1.** Animal-by-animal cerebral blood flow (CBF), cerebrovascular reactivity (CVR), brain volume, and body weight progression with increasing age.

**Figure supplement 2.** Cerebral blood flow (CBF) measurements in cortical (left) and thalamic (right) ROIs.

**Figure supplement 3.** Change in transcutaneously measured $pCO_2$ and respiration profiles during the pseudo-continuous arterial spin labeling (pCASL) measurements.

**Figure supplement 4.** Inversion efficiency values measured at the different time points in cohort 1.

For instance *Chow et al., 2007* and also *Park et al., 2014* used heterozygotic Tg-SwDI mice, whereas in this study, homozygotic Tg-SwDI mice were used. Homozygotic Tg-SwDI mice have been reported to develop more extensive amyloid-β pathology, but with a similar distribution on the micro- and macro-scale in the brain, and similar amyloid-β−40/amyloid-β−42 ratios as hemizygous mice (*Xu et al., 2007*). It seems, however, unlikely that the more severe pathology would result in a reversal of the functional phenotype. Furthermore, the amyloid-β pathology found here (*Figure 7*) is comparable to what is described in the literature for both hemizygous and homozygous mice, namely diffuse parenchymal plaques in the cortex and microvascular accumulation in the thalamus and hippocampus (*Miao et al., 2005*). Second, we used a higher percentage of $CO_2$ for our vascular challenge, that is, 7.5% for 7 min here, versus 5% for 5 min in literature (*Park et al., 2014*). However, it is unlikely that higher $pCO_2$ intake would reverse the phenotype of the Tg-SwDI mice. Furthermore, the $pCO_2$ increase measured here through the skin during U and A is comparable to that reported with blood gas sampling in *Park et al., 2014* (approximately 20 mmHg). Possibly, a lower sensitivity of the transcutaneous $pCO_2$ measurement versus blood gas sampling could explain why our $pCO_2$ increase was not higher. It is also important to mention that in most patient studies (*Dumas et al., 2012*; *van Opstal et al., 2017*) as well as in *Chow et al., 2007* vascular responses to neuronal activity were measured, not baseline perfusion and hypercapnia, which might be differently affected by vascular amyloid-β. However, this is not likely to explain the differences in outcomes, as *Park et al., 2014* showed impaired responses in Tg-SwDI mice to both hypercapnia and neuronal activity. Furthermore, patient studies have also shown baseline perfusion deficits (*van Opstal et al., 2017*). Lastly, in *Chow et al., 2007* and *Park et al., 2014* a craniotomy was performed right before the LDF measurement, whereas here, most of the measurements were performed noninvasively. An attempt was made to account for this difference in experimental set-up by preparing an acute craniotomy in the second cohort of mice. However, even though the brain surface in these animals was visually normal (*Figure 6—figure supplement 1*) after the craniotomy, edema was observed with MRI, and with both imaging modalities marked reductions in CBF and CVR were measured in the underlying brain regions in both WT and Tg-SwDI mice. A likely explanation for the functional impairments after craniotomy is the occurrence of a cortical spreading depression (CSD), which occurs even at minor manipulations on the dura (*Ayata et al., 2004*). In mice specifically, CSDs have been reported to cause a CBF reduction of 40–50%, and enhanced resistance to relaxation by acetylcholine (*Ayata et al., 2004*). It could be hypothesized that the presence of amyloid-β in the Tg-SwDI model enhances the sensitivity of the brain tissue to a CSD. Thus, when measured just after craniotomy, the unnoticed presence of a CSD might give the false idea of direct amyloid-β induced cerebrovascular dysfunction. Interestingly, a recent study also reported that skull removal was necessary to detect cerebrovascular dysfunction in a different mouse model of amyloid-β accumulation, providing some support for this hypothesis (*Sharp et al., 2020*). Nevertheless, because our WT mice showed similar functional deficits as the Tg-SwDI mice, we currently cannot substantiate this hypothesis. It is important to mention that the researcher that performed the craniotomy in this study is highly skilled in this procedure, underlining that this is not merely a matter of experience. Hence, it is of interest to elucidate in future research if our surgical procedure indeed triggers a CSD, and whether Tg-SwDI mice are more susceptible to CSDs.

Alternatively, our results could indicate that the mouse cerebral circulation is less affected by amyloid-β accumulation than the human cerebral circulation. However, other amyloidosis models did

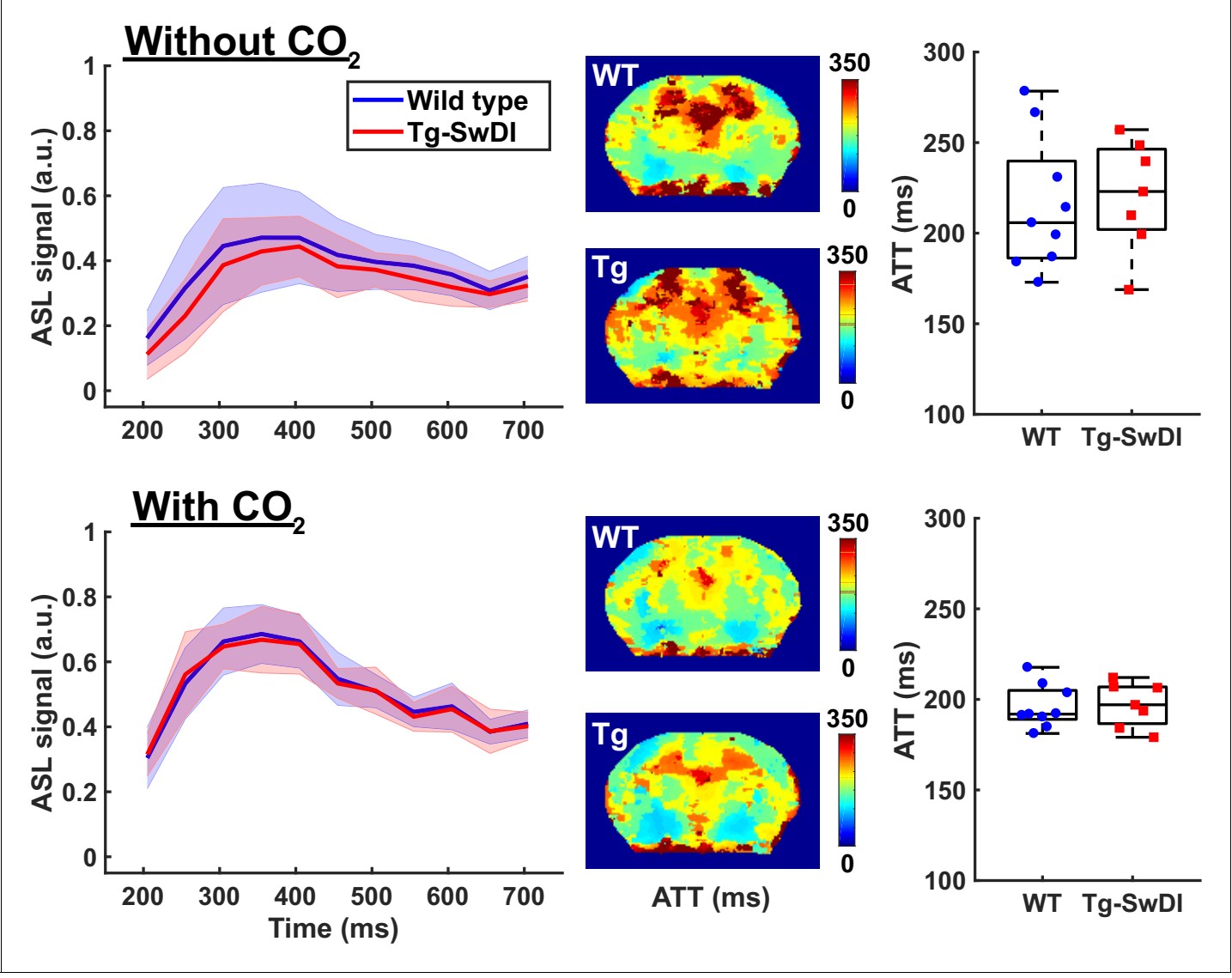

**Figure 4.** Arterial transit time (ATT) measurements acquired in the mid-brain of 12 months old wild-type (WT) and transgenic Swedish Dutch Iowa (Tg-SwDI) mice. On the top row, measurements acquired at baseline are displayed, on the bottom row measurements acquired while administering 7.5% $CO_2$. On the left column, graphs display the measured arterial spin labeling (ASL) signal (mean ± standard deviation) plotted against increasing post-label delay times. In the middle, maps are displayed that show averaged arrival times of the ASL signal. The maps were acquired in a mid-brain slice and are averaged for WT (top) and Tg-SwDI mouse (bottom). On the right, boxplot representations of the ATT values obtained in all mice are displayed, where circles and squares represent individual mice.

show early impairments in cerebrovascular function, even in a noninvasive set-up, such as the APP23 model (*Mueggler et al., 2002*; *Maier et al., 2014*). Because the APP23 model mainly shows arteriolar CAA pathology (*Sturchler-Pierrat et al., 1997*), altogether this might indicate that predominantly CAA pathology on the *arteriolar* side of the vasculature is responsible for the observed CBF and CVR impairments in patients. It is important to note, however, that even without CBF limiting pathology, capillary dysfunction could lead to inefficient oxygen extraction from the capillary network (*Østergaard et al., 2016*). Indeed, hypoxia-induced factor angiopoietin-4 was found to be highly expressed in capCAA patients (*Chakraborty et al., 2018*), indicating that possibly capCAA could induce hypoxic conditions. It would thus be interesting to validate whether similar conditions are found in the Tg-SwDI mouse model.

To the best of our knowledge, this is the first study with ASL-MRI in mice where the CBF responses to hypercapnia under isoflurane and U and A anesthesia were directly compared. Large

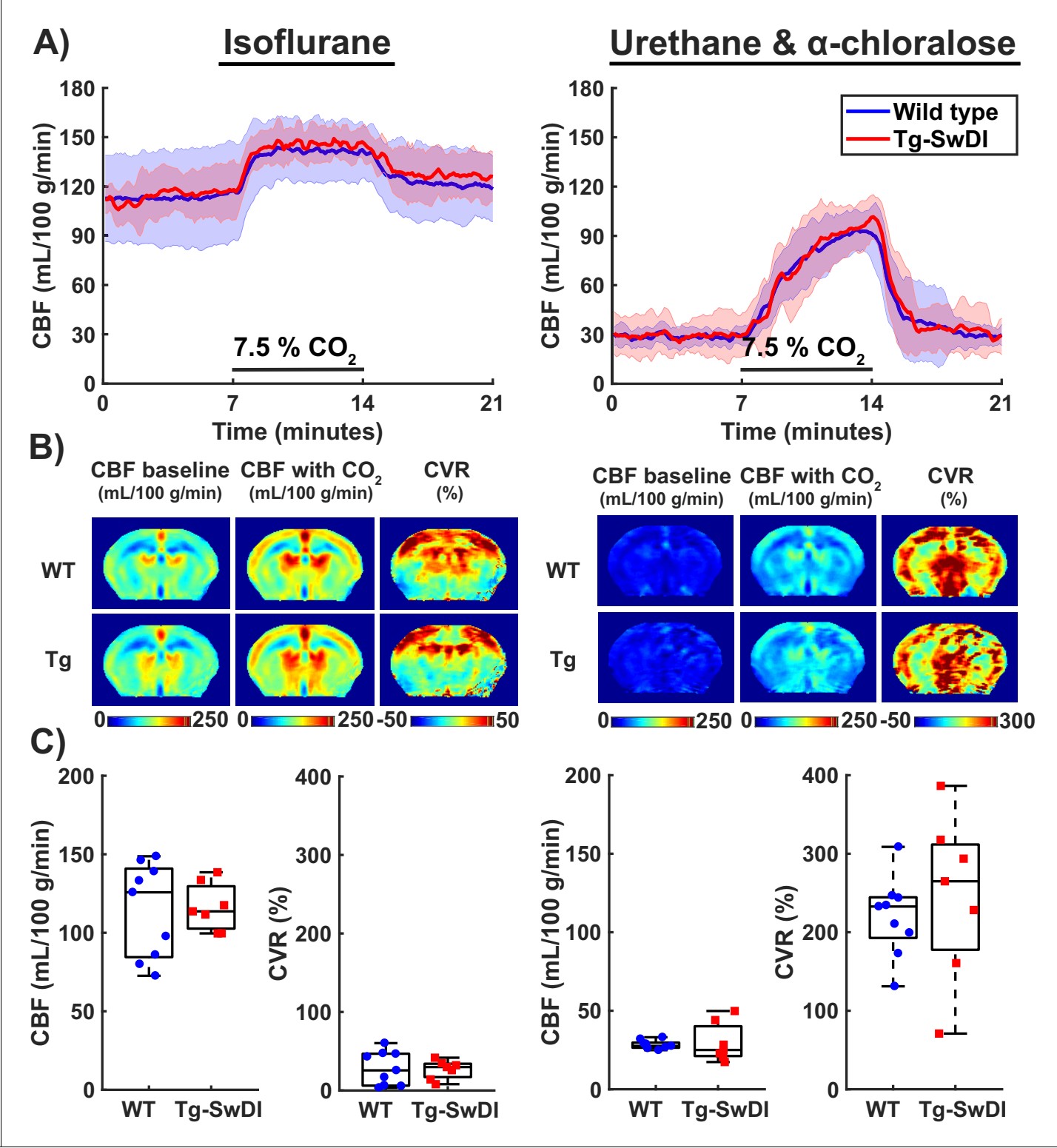

**Figure 5.** Cerebral blood flow (CBF) and cerebrovascular reactivity (CVR) acquired during isoflurane anesthesia and urethane and α-chloralose (U and A) anesthesia. (**A**) 21-min CBF time profiles acquired in the mid-brain in wild-type (WT) and transgenic Swedish Dutch Iowa (Tg-SwDI) mice under either isoflurane anesthesia (left, 12 months old) or U and A anesthesia (right, 10 days later in the same mice). $CO_2$ was administered between minutes 7 and 14. (**B**) Mid-brain CBF and CVR maps averaged for WT (top) and Tg-SwDI (bottom) mice. Note that the CVR maps during U and A anesthesia are scaled differently than the CVR maps under isoflurane due to the marked difference in CVR. (**C**) Boxplot representations of the baseline CBF and CVR group values, where dots and circles represent individual mice.

*Figure 5 continued on next page*

*Figure 5 continued*

The online version of this article includes the following figure supplement(s) for figure 5:

**Figure supplement 1.** Cerebral blood flow (CBF) change observed during switching from isoflurane anesthesia to urethane and α-chloralose (U and A).
**Figure supplement 2.** Change in transcutaneously measured pCO2 and respiration profiles under different anesthesia protocols.

differences in hemodynamic parameters were observed, with the very low CBF during U and A anesthesia as the most striking observation. The low CBF during U and A is non-physiological and thus forms a limitation of this study. Interestingly, U and A is considered one of the more suitable anesthesia protocols for cerebral hemodynamic studies in mice, as this protocol has been reported to have relatively mild hemodynamic effects (*Janssen et al., 2004*; *Wang et al., 2010*) and to maintain cerebral autoregulation (*Dalkara et al., 1995*). However, these studies did not provide absolute CBF estimates, and our results indicate that the stable hemodynamics come together with very low baseline CBF, which may not always be favorable and certainly do not represent normal baseline physiology. It would be of interest to identify the underlying cause for this low CBF.

This is also one of the few studies where a direct comparison was established between hypercapnic CBF responses measured with ASL-MRI versus LDF. The results must, however, be interpreted with care due to the small group size and the ±0.5 hr delay between the two measurements. Both readouts consistently reported preservation of cerebrovascular function in Tg-SwDI mice and a severe reduction of CVR after craniotomy, but CVR measurements with ASL-MRI were nearly twice as high compared to those obtained with LDF. It is unclear where this difference comes from. As the two imaging modalities are sensitive to different blood components (plasma or red blood cells), it could be argued that local hematocrit changes during hypercapnia could explain this difference. Indeed, hematocrit has been reported to decrease during hypercapnia in the rat brain (*Keyeux et al., 1995*), but with only a 10% decrease during a 10% $CO_2$ challenge, which is too low to explain the observed difference. A more convincing explanation could be found in an interesting study where ASL-MRI and LDF were performed simultaneously (*He et al., 2007*) and point in the direction of an underestimation by LDF. The authors showed that by varying the fiber separation distance in the LDF probe, which changes the measured cortical depth, a different CBF response was measured. Consequently, the magnitude of the CBF response was either comparable to that

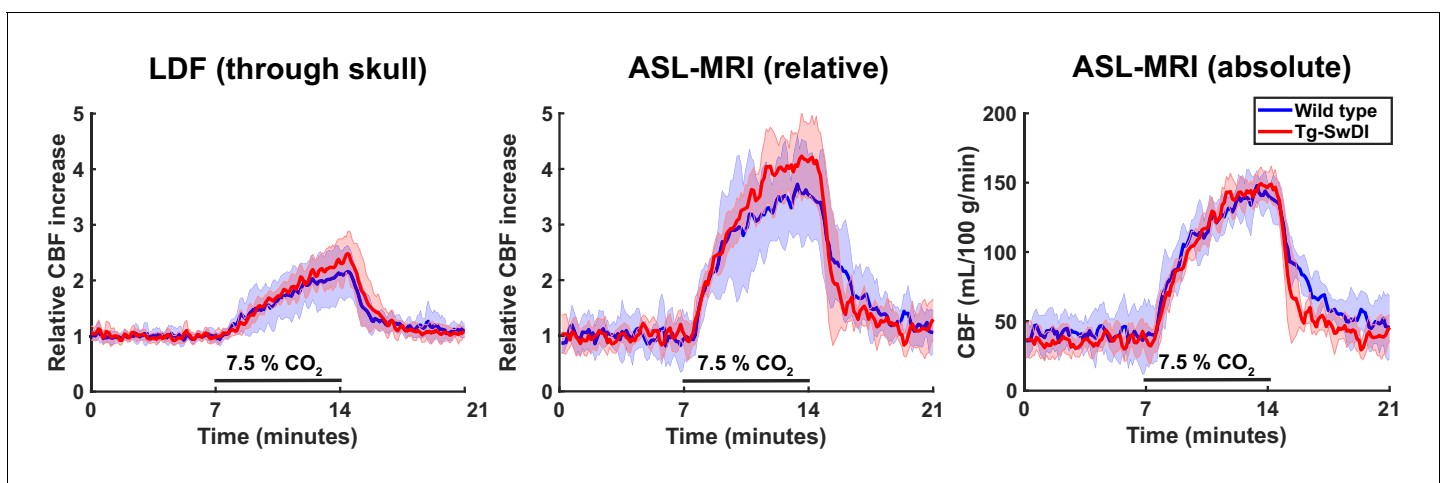

**Figure 6.** Cerebral blood flow (CBF) time profiles acquired with laser Doppler flowmetry (LDF) and arterial spin labeling (ASL)-magnetic resonance imaging (MRI). On the left, 21-min CBF time profiles acquired with LDF in the somatosensory cortex are displayed for wild-type (WT) and transgenic Swedish Dutch Iowa (Tg-SwDI) mice. In the middle, 21-min CBF time profiles are displayed that are acquired with ASL-MRI in the left somatosensory cortex, after baseline correction, so the profiles can be compared to the LDF time profiles. On the right are the same profiles as in the middle, without baseline correction.

The online version of this article includes the following figure supplement(s) for figure 6:

**Figure supplement 1.** Cerebrovascular reactivity (CVR) measurements with laser Doppler flowmetry (LDF) and arterial spin labeling-magnetic resonance imaging ( (ASL-MRI) with and without skull removal.

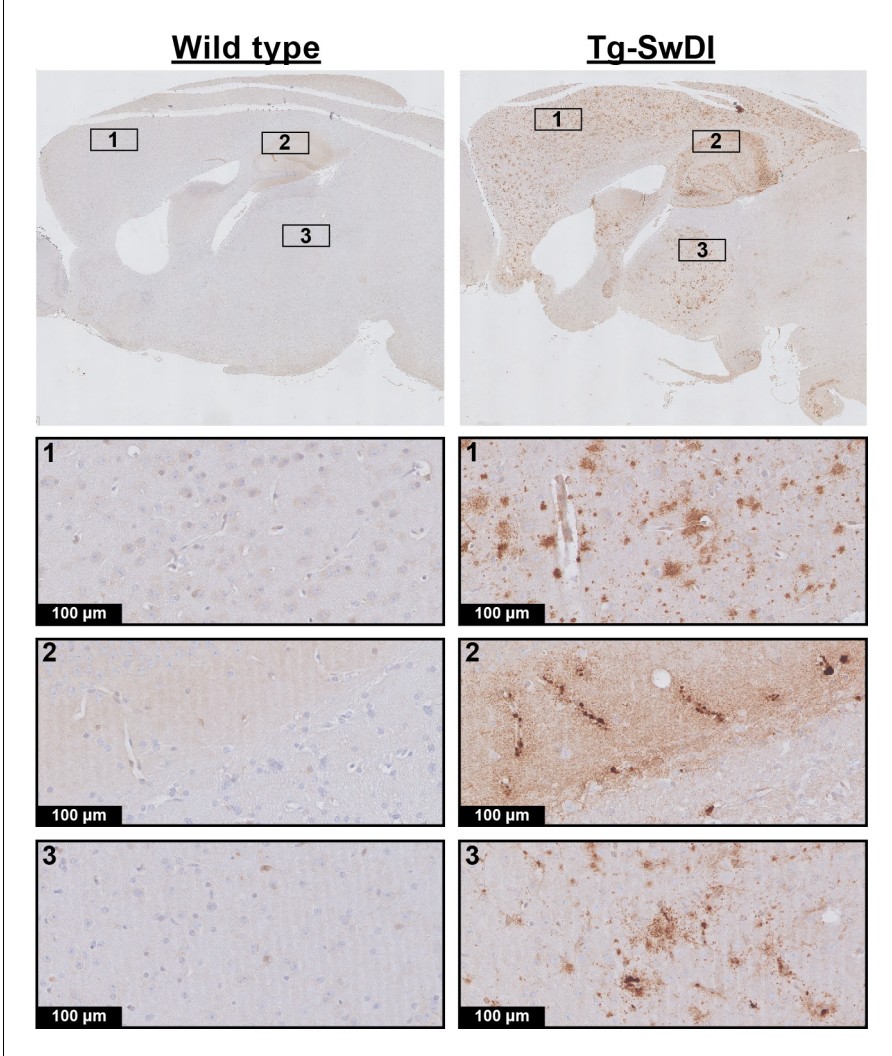

**Figure 7.** Amyloid-β histology. Shown are representative stainings of wild-type (WT) and transgenic Swedish Dutch Iowa (Tg-SwDI) mice of 12.3 months old. The upper row is an overview image, and the other rows are zoomed in regions from the overview image.

measured with ASL-MRI, or up to two times lower, which is comparable to our results. The authors measured CBF responses to electrical whisker stimulation through a thinned skull in rats under urethane anesthesia, and concluded that when larger cortical depths were measured, LDF underestimated the CBF response. It is unclear how these results would exactly translate to our different set-up (smaller cortical thickness in the mouse, intact skull, and different vascular challenge), but indicates that investigating the cortical depth measured with a set-up like ours is a compelling area for future research. Interestingly, the absolute CBF responses measured with ASL-MRI by *He et al., 2007* were comparable to those measured here.

Lastly, another finding in our study that is of interest is the observation of edema and the local reduction in CBF and CVR with MRI after craniotomy. This was observed in all animals, while the brain tissue was normal by visual inspection. Normal appearance of brain tissue is therefore not enough to conclude that the brain tissue is healthy when working with acutely prepared cranial windows.

In conclusion, this study shows that cerebrovascular function in the Tg-SwDI mouse model of microvascular amyloidosis is preserved up to 12 months of age, despite the high amyloid-β burden observed at that age. This observation was confirmed using two different anesthesia protocols and two different imaging modalities. These observations call into question to what extent microvascular

amyloidosis as seen in the Tg-SwDI mouse model is a correct model for cerebrovascular dysfunction as observed in CAA patients, and calls for further research to clear up the discrepancy in results.

# Materials and methods

## Key resources table

| Reagent type (species) or resource | Designation | Source or reference | Identifiers | Additional information |
|---|---|---|---|---|
| Strain, strain background (*Mus musculus*) | C57Bl/6J (WT) | Jackson lab | RRID:IMSR_JAX:000664 | |
| Strain, strain background (*Mus musculus*) | C57BL/6-Tg(Thy1-APPSwDutIowa)-BWevn/Mmjax (Tg-SwDI) | Jackson lab | MMRRC Stock No: 34843-JAX | |
| Antibody | Anti-beta amyloid antibody (rabbit polyclonal) | Abcam | ab2539 RRID:AB_303141 | '(1:1000)' dilution |
| Antibody | Anti-rabbit IgG antibody (pig polyclonal) | Dako | E0431 | '(1:300)' dilution |
| Ahemical compound, drug | α-Chloralose | Sigma-Aldrich | C0128 | 50 mg/kg |
| Chemical compound, drug | Urethane | Sigma-Aldrich | U2500 | 750 mg/kg |
| Software, algorithm | DASYLab | National Instruments | https://www.mccdaq.com/DASYLab-Resources.aspx | Version 13 |
| Software, algorithm | MATLAB | Mathworks | RRID:SCR_001622 | |
| Software, algorithm | SPSS | IBM | RRID:SCR_002865 | Version 26 |
| Software, algorithm | EVolution | Denis de Senneville et al. | Phys Med Biol 2016 | |
| Software, algorithm | MIA / MP3 | Brossard et al. | Front Neuroinform 2020 | |
| Software, algorithm | G*Power | Faul et al. | Behav Res Methods 2007 | |
| Other | Avidin-Biotin Complex kit | Vector Laboratories | Vectastain | |

## Animals

All the experiments were approved by the local ethics committee ('Leiden University Medical Center Instantie voor Dierenwelzijn') and the national ethical committees ('Centrale Commissie Dierproeven') under OZP PE.18.029.002 of AVD116002017859, and the experiments have been reported in compliance with the ARRIVE guidelines (*Kilkenny et al., 2010*).

Two cohorts of homozygous Tg-SwDI mice and age- and gender-matched WT controls on a C57Bl/6J background were ordered from the Jackson Laboratory (Bar Harbor, ME, USA). The first cohort was used for a hypothesis-driven study. Based on a previous report by *Park et al., 2014*, we expected an effect size of 30 percentage points difference – in cortical CBF response to a hypercapnia challenge – between Tg-SwDI and WT mice, and a standard deviation of 12% in both groups. With a power calculation done with G*Power software (*Faul et al., 2007*), where we assumed an alpha of 0.05 and a power of 80% for a two-sided Mann–Whitney U-test, we estimated four animals would be needed to detect a significant difference in cortical CVR. However, to counteract the

possibility of a lower effect size, higher standard deviation and/or animal drop-out, we chose to increase our sample size with five animals per group, thus to nine animals per group in total. The cohort consisted of all male mice which were followed longitudinally, that is, they were imaged at 3, 6, 9, 12, and 12.3 months of age. The second cohort was used for an exploratory study and consisted of four Tg-SwDI (two females) and four WT mice (two females) which were imaged once at 8 months of age. No blinding was performed when the mice were imaged. Two of the transgenic mice were taken out of the longitudinal study, which was not related to the study itself: the first mouse was severely wounded by a cage mate (at 2.5 months of age), the other due to ulcerative dermatitis (at 10 months of age). The mice in the longitudinal cohort were initially co-housed; however, after the aggression incident at 2.5 months, all mice were housed individually to prevent further aggression and to keep the conditions between the mice comparable. Two study-related dropouts occurred in the second cohort (one WT and one transgenic) due to failure of the intubation procedure (see below). Housing consisted of individually ventilated cages in rooms with a 12 hr day/night rhythm. The cages were supplied with cage bedding, cage enrichment (a small roll to hide in and a block of wood), and unlimited chow food and water. The body weights of the mice were measured monthly.

Animal preparation – the isoflurane anesthesia protocol during the first four imaging sessions in cohort 1 was similar to the 'low isoflurane protocol' as described in *Munting et al., 2019*. In this protocol, the isoflurane concentration is kept low both during induction (2.0% for 5 min) and maintenance (1.25%), to limit the vasodilatory effects of the anesthetic, which would impact the reactivity measurements. The mouse was freely breathing during the scan. During the fifth imaging session, anesthesia was maintained using 750 mg/kg urethane and 50 mg/kg alpha-chloralose (U and A), similar to the concentrations used by *Park et al., 2014*. Anesthesia induction with U and A was performed using isoflurane (3.5%), which was thereafter decreased to 1.75%. Then, the trachea of the animal was surgically intubated for mechanical ventilation at 80 bpm, 25% inspiration rate, and 1.7 psi (MRI-1 ventilator, CWE Inc, USA). Additionally, an i.p. catheter was placed to administer U and A anesthesia during the MRI scans. Thirty-five minutes after U and A injection, the isoflurane was decreased over the course of 10 min to 0%. During all sessions, transcutaneous partial pressure of carbon dioxide (tc-pCO$_2$) was monitored (TCM Radiometer, Denmark) with a neonate probe attached to the previously shaved skin on the right flank of the mouse (*Ramos-Cabrer et al., 2005*). Breathing rates were monitored using a pressure-sensitive pad placed below the animal (SA Instruments, NY, USA). Temperature was maintained at 36.5°C using a feedback-controlled waterbed with rectal probe (Medres, Germany) and the head was stabilized with a bite bar and ear bars.

The second cohort was also anesthetized with U and A; however, in this cohort, the i.p. injection was given directly after the induction with 3.5% isoflurane. Again, isoflurane was kept at 1.75% for 35 min after injection, after which it was decreased to 0% over the course of 10 min. While still at 1.75% isoflurane, the trachea was surgically intubated for mechanical ventilation and additionally, a craniotomy of 3 mm in diameter was prepared over the right somatosensory cortex. The dura mater was left intact and kept moist with sterile PBS. Similar physiological monitoring was performed as in the first cohort. Fifteen minutes after the isoflurane reached 0%, the LDF recording was started. After the LDF measurement, and before the animal was placed in the MRI scanner, the skin was placed back over the exposed skull and brain and sutured to limit susceptibility artifacts.

## Image acquisition

MRI acquisition – a 7 T Pharmascan MRI scanner (Bruker, Germany) with a 23 mm transmit-receive volume coil was used. After proper placement of the mouse head in the bore was confirmed with a scout scan, three standard Bruker T2-weighted RARE scans (TE/TR = 35.0 ms/2500 ms; 78 × 78 × 700 µm$^3$ resolution) were performed in all three directions for consistent planning across animals of the subsequent pseudo-continuous ASL (pCASL) scans. The coronal slice package with 21 slices was additionally used for quantification of the brain volume. The pCASL scan protocol was similar as in *Munting et al., 2019*. In short, the phase of the pCASL labeling was first optimized using pre-scans (*Hirschler et al., 2018b*). Thereafter, CBF and CVR were measured using a 21-min pCASL scan with 180 dynamics, a labeling duration (τ) of 3000 ms, a post-labeling delay (PLD) of 300 ms, and a five slice spin-echo echo planar imaging (SE-EPI) readout with 225 µm$^2$ resolution and 1.5 mm slice thickness (no slice gap). 7.5% CO$_2$ was administered to the mouse from minute 7 till minute 14. For the last imaging session (12.3 months) of the mice in the first cohort, the number of dynamics of

the pCASL scan was extended to 767 (scan duration of 90 min) to also capture the effect of switching from isoflurane to U and A anesthesia on CBF. Specifically, the following steps were performed during the 1.5 hr ASL scan: at 5 min, U and A was injected, between minutes 40 and 50 isoflurane was decreased from 1.75% till 0% and between minutes 72 and 79, 7.5% $CO_2$ was administered. At the fourth scan session (12 months), two time-encoded pCASL (te-pCASL) sequences were additionally acquired per mouse for measuring the ATT. The second te-pCASL sequence was performed while administering 7.5% $CO_2$ to the mouse, which allowed measuring the effect of arterial $pCO_2$ elevation on the ATT. The scan parameters of the te-pCASL sequence were the same as in *Hirschler et al., 2018a*; *Hirschler et al., 2018b* except for the resolution, which was decreased from 225 to 337 $\mu m^2$ to increase the signal-to-noise ratio (SNR). A three slice SE-EPI readout was used, with the same slice orientation as slices 1, 3, and 5 of the standard pCASL scan (thus with a slice gap of 1.5 mm). During every imaging session, for CBF quantification purposes, the $T_1$ of the tissue ($T_{1t}$) and the tissue magnetization ($M_{0t}$) were estimated by collecting an additional inversion recovery scan with the same five slice SE-EPI readout as the pCASL scans. Furthermore, a pCASL flow-compensated FLASH was acquired at the level of the carotids, 3 mm downstream of the labeling plane, to measure the labeling efficiency ($\alpha$). An additional $T_2$-weighted (T2W) RARE anatomical sequence was acquired with the same slice orientation as that of the pCASL scan for registration purposes.

LDF acquisition – A PeriFlux system with a PF 5010 LDPM unit and two Laser Doppler probes was used for LDF monitoring (Perimed Instruments, Sweden). Both probes were placed at the somatosensory cortex, one directly on the exposed skull above the left hemisphere, the other approximately 0.5 mm above the exposed brain tissue of the right hemisphere (hereafter referred to as the left and right probe, respectively). Extraction of the LDF signal profile was done through DASYLab software (version 13, National Instruments, Germany).

After the last MRI measurement, a subgroup of mice (6 Tg-SwDI and 6 WT) was i.v. injected with 200 μL of DyLight-594-coupled lectin (lycopersicum esculentum, VectorLabs, CA, USA) in the tail vein for staining of the endothelium. This was followed 3 min later by an i.p. overdose of pentobarbital after which the mouse was transcardially perfused with 20 mL of ice-cold PBS and 20 mL of ice-cold 4% PFA. The brain was isolated and fixed overnight in 4% PFA. The tissue was thereafter preserved at 4°C in PBS with 0.02% sodium azide until further processing. The other mice followed the exact same steps, but without the i.v. lectin injection.

## Image processing

T2W processing – One full-brain volume of interest (VOI) was manually drawn on one of the 21-slice T2W RARE scans, after which the VOI was propagated to the T2W RARE scans of the other data sets (both to other mice and to other time points) using the EVolution algorithm (*Denis de Senneville et al., 2016*). To derive the brain volume, the number of voxels in the VOI was multiplied by the voxel volume.

ASL image processing – The ASL image processing pipeline was the same as used in *Munting et al., 2019*. In short, the SE-EPI frames within one pCASL scan were aligned using the image processing toolbox of MATLAB (version 2018b, Mathworks, USA). Subsequently CBF, CVR, and ATT were calculated using the MATLAB-based 'Multi-Image Analysis (MIA)' software developed at the Grenoble Institute of Neuroscience (Grenoble, France) (*Brossard et al., 2020*). CBF was calculated for each pair of label/control images, expressed in mL/100 g/min, and derived using Buxton's general kinetic perfusion model (*Buxton et al., 1998*) with the following equation:

$$CBF = \frac{\lambda \cdot \Delta M \cdot \exp(PLD/T_{1b})}{2 \cdot \alpha \cdot T_{1t} \cdot M_{0t} \cdot (1 - \exp(-\tau/T_{1t}))}$$

where $\lambda$ is the blood–brain partition coefficient, that is, 0.9 mL/g (*Herscovitch and Raichle, 1985*), $\Delta M$ is the signal difference of the label and control images from the standard pCASL scans and $T_{1b}$ is the longitudinal relaxation time of blood, that is, 2230 ms at 7 T (*Dobre et al., 2007*). Baseline CBF was defined as the average of the last 20 repetitions ($\approx$ 2.3 min) before the start of $CO_2$ administration. CVR was defined as the ratio calculated from the average of the last 20 repetitions during $CO_2$, over the average of the last 20 repetitions before $CO_2$ administration. To calculate ATT, the decoded signal from the te-pCASL scans was used, as described in *Hirschler et al., 2018a*. Cortical, thalamic, and full mid-brain brain (ROIs) were manually drawn on one of the T2W RARE scans, after

which they were propagated to the T2W RARE scans of the other data sets (both to other mice and to other time points) using the EVolution algorithm (*Denis de Senneville et al., 2016*). This image registration method was also applied to position the ROIs in the corresponding SE-EPIs. This allowed to retrieve CBF, CVR, and ATT values for different brain regions. CBF time profiles were all filtered using a sliding window of three time points (after CBF calculation), besides the profiles acquired in cohort 2 in the cortical ROI of the hemisphere where the skull was removed (*Figure 5—figure supplement 1*), which were filtered with a sliding window of seven time points.

LDF processing – The LDF signal time profiles extracted with DasyLab were filtered with a sliding window of three time points and normalized to the average signal during the first 7 min of the measurement (baseline signal).

## Immunohistochemistry (IHC)

The fixed brains were embedded in paraffin, and subsequently cut in sections of 5 µm. After deparaffinization with xylene and rehydration through graded ethanol series, the slides were cooked for 20 min in citrate buffer for antigen retrieval. Slides were then stained overnight at 4°C with anti-amyloid-β antibody (1:1000; Abcam ab2539) followed by a 1 hr room temperature incubation with biotinylated secondary antibody (1:300; Dako E0431). Immunodetection was visualized using an Avidin-Biotin Complex kit (Vector Laboratories, UK), and sections were counterstained with haematoxylin before mounting. The slides were digitized with an automatic bright field microscope (Philips Ultra Fast Scanner, Philips, the Netherlands) and assessed by one examiner (LPM) for positivity for amyloid-β.

## Statistical testing

To test the effect of genotype on CBF, CVR, ATT, and tc-pCO$_2$, Mann–Whitney U-tests were performed. For the longitudinal cohort, Friedman tests were performed to test the effect of age on CBF and CVR, post-hoc followed by Wilcoxon signed-rank tests to determine which of the individual age groups differed from each other. Only consecutive age groups were compared to each other to restrict the stringency of the Bonferroni correction for multiple comparisons. Wilcoxon signed-rank tests were used to test the effect of anesthesia on CBF and CVR and to test the effect of CO$_2$ on the ATT. No statistical testing was performed in cohort 2, given the small group size. All tests were performed in the SPSS statistics software package, version 26 (IBM, Armonk, NY, USA). The results of all tests are summarized in *Supplementary file 1*.

## Acknowledgements

The authors would like to thank the following persons: Thas Phisonkunkasem, Nico Jansen, Maarten Schenke, and Else Tolner for their help with setting up the LDF measurements, Laibaik Park for his advice on the urethane and α-chloralose anesthesia protocol, and Ingrid Hegemann-Klein and Marjolein Bulk for their help with the histology. L van der Weerd, E Suidgeest, M Derieppe, and LP Munting are supported by the Netherlands Organization for Scientific Research (NWO) Innovational Research Incentives Scheme (VIDI grant 864.13.014) and the Heart Brain Connection consortium, which is supported by the Netherlands CardioVascular Research Initiative: the Dutch Heart Foundation (CVON 2012–06 HBC), the Netherlands Organisation for Health Research and Development, and the Royal Netherlands Academy of Sciences. MJP van Osch and L Hirschler are supported by the Division Applied and Engineering Sciences of the NWO (VICI grant 016.160.351).

## Additional information

### Funding

| Funder | Grant reference number | Author |
| --- | --- | --- |
| Nederlandse Organisatie voor Wetenschappelijk Onderzoek | 864.13.014 | Leon P Munting<br>Marc Derieppe<br>Ernst Suidgeest<br>Louise van der Weerd |
| Nederlandse Organisatie voor | 016.160.351 | Lydiane Hirschler |

| | | |
|---|---|---|
| Wetenschappelijk Onderzoek | | Matthias van Osch |
| Hartstichting | CVON 2012-06 HBC | Marc Derieppe<br>Ernst Suidgeest |

The funders had no role in study design, data collection and interpretation, or the decision to submit the work for publication.

## Author contributions

Leon P Munting, Conceptualization, Software, Formal analysis, Investigation, Visualization, Methodology, Writing - original draft, Writing - review and editing; Marc Derieppe, Software, Formal analysis, Investigation, Methodology, Writing - review and editing; Ernst Suidgeest, Methodology; Lydiane Hirschler, Baudouin Denis de Senneville, Software, Formal analysis; Matthias JP van Osch, Conceptualization, Supervision, Writing - review and editing; Louise van der Weerd, Conceptualization, Resources, Supervision, Funding acquisition, Project administration, Writing - review and editing

## Author ORCIDs

Leon P Munting ![iD] https://orcid.org/0000-0002-2352-4451
Marc Derieppe ![iD] https://orcid.org/0000-0002-9099-1746
Ernst Suidgeest ![iD] https://orcid.org/0000-0001-8015-9238
Lydiane Hirschler ![iD] https://orcid.org/0000-0003-2379-0861
Matthias JP van Osch ![iD] https://orcid.org/0000-0001-7034-8959
Baudouin Denis de Senneville ![iD] https://orcid.org/0000-0001-5284-8474
Louise van der Weerd ![iD] https://orcid.org/0000-0002-5997-2125

## Ethics

Animal experimentation: All the experiments were approved by the local ethics committee ("Leiden University Medical Center Instantie voor Dierenwelzijn") and the Dutch national ethical committees ("Centrale Commissie Dierproeven") under OZP PE.18.029.002 of AVD116002017859.

## Decision letter and Author response

Decision letter https://doi.org/10.7554/eLife.61279.sa1
Author response https://doi.org/10.7554/eLife.61279.sa2

# Additional files

## Supplementary files

• Supplementary file 1. In this table, the outcomes of the statistical tests are detailed. The tests are grouped per effect measured, that is, genotype effects on tab 1, age effects on tab 2, anesthesia effects on tab 3, and $CO_2$ effects on tab 4. Tests that returned significant effects are highlighted in green.

• Transparent reporting form

## Data availability

Imaging data have been deposited in the EASY online archiving system, and is accessible under: https://doi.org/10.17026/dans-znx-cvf3.

The following dataset was generated:

| Author(s) | Year | Dataset title | Dataset URL | Database and Identifier |
|---|---|---|---|---|
| Munting LP | 2020 | Tg-SwDI ASL-MRI study | https://doi.org/10.17026/dans-znx-cvf3 | ASL-MRI, 10.17026/dans-znx-cvf3 |

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
