## [Decision Letter]

**Acceptance summary:**

In this longitudinal experimental study using the Tg-SwDI mouse model of Alzheimer's disease which develop cerebral amyloid angiopathy, the authors employed arterial spin labeling-MRI longitudinally and laser Doppler flowmetry under anesthesia to measure whether amyloid-β deposits impact the hemodynamic response under hypercapnic challenge. In contrast to the previous reports, Tg-SwDI AD mice and age-matched controls show similar baseline perfusion and cerebrovascular reactivity, with differences in hemodynamic responses demonstrated for age and anesthesia. This work demonstrates that cerebrovascular function remains unaffected by microvascular amyloidosis in the Tg-SwDI AD model and raises the question whether this model is representative of cerebrovascular dysfunction observed in patients with Alzheimer's disease and cerebral amyloid angiopathy.

**Decision letter after peer review:**

Thank you for submitting your article "Cerebral Blood Flow and Cerebrovascular Reactivity Are Preserved in a Mouse Model of Cerebral Microvascular Amyloidosis" for consideration by *eLife*. Your article has been reviewed by two peer reviewers, and the evaluation has been overseen by a Reviewing Editor and a Senior Editor. The following individual involved in review of your submission has agreed to reveal their identity: Philipp Boehm-Sturm (Reviewer #2).

The reviewers have discussed the reviews with one another and the Reviewing Editor has drafted this decision to help you prepare a revised submission.

Summary:

In their manuscript, Derieppe and co-authors report on a longitudinal study of the Tg-SwDI mouse model of cerebral microvascular amyloid-β deposits. The authors employ ASL longitudinally under two regimes of anesthesia (isoflurane or urethane and 𝛼-chloralose) as well as LDF in one time point under isoflurane to measure whether deposits impact the hemodynamic response with CBF and CVR under hypercapnic challenge. In contrast to previous reports, Tg-SwDI and age-matched controls show similar baseline perfusion and cerebrovascular reactivity, with difference in hemodynamic responses demonstrated for age and anesthesia. The authors carefully carried out a comprehensive study to estimate the impact of amyloid-β deposits and the lack of difference between the experimental groups suggests that perhaps the Tg-SwDI may not be an effective model of CAA. While raising possible causes for the discrepancy between the current results and prior reports, the failure to replicate remains unresolved. A strong aspect of the work is the characterization of cerebrovascular reactivity as a function of age using ASL as well as the comparison of anesthesia regimes and ASL to LDF.

Essential revisions:

1) The wild type group demonstrates variability that seems to be greater relative to Tg-SwDI and bimodal at 12 months of age (Figure 3, Figure 5C, and Figure 2—figure supplement 1 in both ROIs). This raises the question of whether indeed the control group is homogenous relative to the transgenic group. Was the physiological state of the animals at 12 months assessed? What were the weights of the animals?

2) Did the authors acquire whole-brain T2W scans? If yes, it will useful to quantify brain volumes as a function of age in both groups to help rule out issues with the control group.

3) Regional effects: The authors report in Figure 2—figure supplement 1 two ROIs which do not show a difference. Is this lack of difference also shown in other ages? It will be useful to present all ages. Further, it is unclear that these ROIs provide enough granulation. Did the authors attempt a direct group-wise voxel-by-voxel comparison of the aligned images to see whether any differences are present?

4) Statistics: A priori sample size calculations were not performed. Although the effect sizes from Park et al. seem extraordinarily high (probably Cohen d>1, stats not well described in the original paper), the present study has low n and is underpowered for a hypothesis-driven study. The statistical analyses, exclusion criteria etc. are well described but were not preregistered and strategies to reduce bias (randomization/blinding) neglected. Please state your hypotheses in the Introduction more clearly. Clearly label your study as exploratory or hypothesis-driven.

5) U&A anesthesia: Animals under U&A anesthesia have extremely low CBF, the animals seem to be in a non-physiological state and all experiments under U&A remain with a question mark. Although this is an important finding also for the previous investigations of Park and others, this presents a major limitation of the study.

6) Craniotomy: Instead of LDF, laser speckle without craniotomy may have been the method of choice to corroborate the MRI results. The craniotomy is correctly discussed as a major confounder. This puts all the results of cohort 2 into question and weakens the study's main finding. Almost any manipulation on the dura induces a spreading depolarization (SD) in mice if you are not very well aware of it (i.e. measure it) and highly trained. Most importantly, the first SD leads to a massive reduction of cortical CBF >40% for >2 h which is well described and very specific to mice (in contrast to rats/pigs/humans, https://pubmed.ncbi.nlm.nih.gov/15529018/). The problem is, that after this baseline drop, the vascular response to SD looks normal in the mouse but essentially isn't. The implications of the phenomenon for a CO2 challenge/CVR have unfortunately not yet been investigated. For a craniotomy on the right side, the phenomenon is observed on the right hemisphere.

Is this lateral effect present in the data? It is unfortunate, that the two confounding factors (different anesthesia and craniotomy) cannot fully be unmixed, the missing experiment is LDF under isoflurane (plus necessary analgesia).

---

## [Author Response]

Essential revisions:1) The wild type group demonstrates variability that seems to be greater relative to Tg-SwDI and bimodal at 12 months of age (Figure 3, Figure 5C, and Figure 2—figure supplement 1 in both ROIs). This raises the question of whether indeed the control group is homogenous relative to the transgenic group. Was the physiological state of the animals at 12 months assessed? What were the weights of the animals?

We thank the reviewer for the observation of the greater variability and bimodality of the perfusion in the wild type animals at 12 months of age. We agree that an explanation for this phenomenon would make the data stronger. For that reason, as well as to further support the data in general, we decided to add data on body weight, brain volume and respiration rate to the manuscript (see Figure 3—figure supplement 1, Figure 3—figure supplement 3 and Figure 5—figure supplement 2). However, when looking at the added curves, there is no clear indication that either body weight, brain volume or respiration could explain the bimodality in the cerebral blood flow (CBF) of the wild type mice at 12 months of age. To be sure, we performed a correlation analysis of the CBF in wild type mice at 12 months of age, with body weight, brain volume and respiration.

The correlation analysis illustrates that indeed, there is no indication that either of the three parameters could explain the bimodality in the CBF of the wild type mice. We considered the correlation analysis too detailed for the manuscript, so we have not added this to the manuscript. However, we explicitly mentioned the greater variability in the WT group in the Results section.

2) Did the authors acquire whole-brain T2W scans? If yes, it will useful to quantify brain volumes as a function of age in both groups to help rule out issues with the control group.

We did acquire T2W scans, and we agree with the reviewer that adding brain volume data would make the data stronger. In the Materials and methods section, we have therefore added the following description of the processing of the T2W scans:

“T2W processing – One full-brain volume of interest (VOI) was manually drawn on one of the 21-slice T2W RARE scans, after which the VOI was propagated to the T2W RARE scans of the other datasets (both to other mice and to other time points) using the EVolution algorithm (Denis de Senneville et al., 2016). To derive the brain volume, the number of voxels in the VOI was multiplied by the voxel volume.”

The results of the brain volume calculation have been added in Figure 3—figure supplement 1 (every colored line is an individual mouse, and the black dashed line is the group average). As illustrated in Author response image 1, brain volume did not provide an explanation for the bimodality in the wild type group. Furthermore, no brain volume differences were observed between wild type and transgenic animals. This observation is also referred to in the Results section.

3) Regional effects: The authors report in Figure 2—figure supplement 1 two ROIs which do not show a difference. Is this lack of difference also shown in other ages? It will be useful to present all ages. Further, it is unclear that these ROIs provide enough granulation. Did the authors attempt a direct group-wise voxel-by-voxel comparison of the aligned images to see whether any differences are present?

We agree with the reviewer that more information on the regional analysis is needed. Therefore, as suggested, we added the information on the regional analysis for the other time points in Figure 3—figure supplement 2. Note that, to not overcrowd this figure, we decided to only show the boxplot analysis, not the CBF time-profiles.

We also agree with the reviewer that using the aligned images to add group-wise maps for more granulation would make the data stronger. We therefore decided to do this for all time points. Because we were positively surprised about the quality of the group-wise aligned CBF and CVR maps, we decided to add a new figure, which fully focuses on these new maps (Figure 2). However, no differences were observed between the wild type and transgenic CBF and CVR maps. Because of the much better quality of the group average maps versus the example maps, we decided to also replace the example maps in Figure 4 and Figure 5 with group average maps (not added in this response letter). Furthermore, we realized that in Figure 4, we had previously used a different slice than in the other figures (posterior slice instead of mid-brain slice), so we updated Figure 4 with mid‑brain data so it is in line with the rest of the data. Please also note that, because of the addition of the new Figure 2, the maps in former Figure 2 became redundant. Therefore, we decided to merge the former Figure 2 (without the maps) and former Figure 3 together in a new Figure 3.

4) Statistics: A priori sample size calculations were not performed. Although the effect sizes from Park et al. seem extraordinarily high (probably Cohen d>1, stats not well described in the original paper), the present study has low n and is underpowered for a hypothesis-driven study. The statistical analyses, exclusion criteria etc. are well described but were not preregistered and strategies to reduce bias (randomization/blinding) neglected. Please state your hypotheses in the Introduction more clearly. Clearly label your study as exploratory or hypothesis-driven.

Indeed, we did not report our a priori sample size calculations, and agree with the reviewer that they contain valuable information and therefore should be added. We had already performed these calculations when applying for the animal experimentation license and now added these calculations with a description in the manuscript:

“The first cohort was used for a hypothesis-driven study. Based on a previous report by Park et al., 2014, we expected an effect size of 30 percentage points difference – in cortical CBF response to a hypercapnia challenge – between Tg-SwDI and wild type mice, and a standard deviation of 12 % in both groups. […] However, we would also like to point out that during analysis, blinding was performed, and randomization is not applicable to this study, as the animals did not undergo treatment.”

5) U&A anesthesia: Animals under U&A anesthesia have extremely low CBF, the animals seem to be in a non-physiological state and all experiments under U&A remain with a question mark. Although this is an important finding also for the previous investigations of Park and others, this presents a major limitation of the study.

We agree with the reviewer that the low CBF during U&A anesthesia is questionable, and represents a non-physiological state. We therefore emphasize this in the Discussion:

“The low CBF during U&A is non-physiological and thus forms a limitation of this study.”

6) Craniotomy: Instead of LDF, laser speckle without craniotomy may have been the method of choice to corroborate the MRI results. The craniotomy is correctly discussed as a major confounder. This puts all the results of cohort 2 into question and weakens the study's main finding. Almost any manipulation on the dura induces a spreading depolarization (SD) in mice if you are not very well aware of it (i.e. measure it) and highly trained. Most importantly, the first SD leads to a massive reduction of cortical CBF >40% for >2 h which is well described and very specific to mice (in contrast to rats/pigs/humans, https://pubmed.ncbi.nlm.nih.gov/15529018/). The problem is, that after this baseline drop, the vascular response to SD looks normal in the mouse but essentially isn't. The implications of the phenomenon for a CO2 challenge/CVR have unfortunately not yet been investigated. For a craniotomy on the right side, the phenomenon is observed on the right hemisphere.Is this lateral effect present in the data? It is unfortunate, that the two confounding factors (different anesthesia and craniotomy) cannot fully be unmixed, the missing experiment is LDF under isoflurane (plus necessary analgesia).

We thank the reviewer for addressing the possible confounding effect of a spreading depolarization in the mouse cortex after skull removal. We agree with the reviewer that this is a possible explanation for the altered CBF and CVR we observed after craniotomy, and we have added and referenced this idea in the Discussion:

“However, even though the brain surface in these animals was visually normal (Figure 6—figure supplement 1) after the craniotomy, edema was observed with MRI, and with both imaging modalities marked reductions in CBF and CVR were measured in the underlying brain regions in both WT and Tg‑SwDI mice. […] Hence, it is of interest to elucidate in future research if our surgical procedure indeed triggers a CSD, and whether Tg‑SwDI mice are more susceptible to CSDs.”

Because with ASL-MRI, we observe a significant reduction in baseline cortical blood flow in the hemisphere where the craniotomy was performed, our data confirms the notion of the reviewer (“cortical CBF >40%”, see Figure 6—figure supplement 1). Furthermore, we also observe a profound reduction in CVR to hypercapnia in the same brain region, which can be seen in the same figure.

Because we used two LDF probes – one of the probes measured CBF through the skull in the untouched hemisphere – we could see, as the reviewer points out, a lateral effect in the data (see Figure 6—figure supplement 1). Furthermore, this lateral effect was confirmed with ASL-MRI (see same figure). We are therefore convinced that the impaired response after craniotomy is not an effect of anesthesia, but an effect of the craniotomy. In addition, our ASL-MRI data in the unaffected hemisphere corresponded well with our measurements under U&A anesthesia in cohort 1, thus further supporting this notion. Therefore, we did not acquire additional data with LDF under isoflurane. Because we did not write this clearly before, we did now emphasize in the Results and in the figure legends, that we measured LDF with two probes in two hemispheres, both through the skull and after craniotomy.